# The Implication of Gastric Microbiome in the Treatment of Gastric Cancer

**DOI:** 10.3390/cancers14082039

**Published:** 2022-04-18

**Authors:** George Pappas-Gogos, Kostas Tepelenis, Fotis Fousekis, Konstantinos Katsanos, Michail Pitiakoudis, Konstantinos Vlachos

**Affiliations:** 1Department of Surgery, University Hospital of Ioannina, 45500 Ioannina, Greece; kostastepelenis@gmail.com (K.T.); kvlachos@uoi.gr (K.V.); 2Department of Gastroenterology, University Hospital of Ioannina, 45500 Ioannina, Greece; fotisfous@gmail.com (F.F.); khkostas@hotmail.com (K.K.); 3Department of Surgery, University Hospital of Alexandroupolis, 68100 Alexandroupolis, Greece; mpitiak@med.duth.gr

**Keywords:** gastric cancer, gastric microbiome, chemotherapy, immunotherapy

## Abstract

**Simple Summary:**

Gastric cancer (GC) represents the fifth most common cancer worldwide. Recent developments in PCR and metagenomics clarify that the stomach contains a powerful microbiota. Conventional treatments for GC that include surgery, chemotherapy, and radiotherapy are not very effective. That’s why new therapeutic strategies are needed. The intestinal microbiota is involved in oncogenesis and cancer prevention, and the effectiveness of chemotherapy. Recent studies have shown that certain bacteria may enhance the effect of some traditional antineoplastic drugs and immunotherapies.

**Abstract:**

Gastric cancer (GC) is one of the most common and deadly malignancies worldwide. Helicobacter pylori have been documented as a risk factor for GC. The development of sequencing technology has broadened the knowledge of the gastric microbiome, which is essential in maintaining homeostasis. Recent studies have demonstrated the involvement of the gastric microbiome in the development of GC. Therefore, the elucidation of the mechanism by which the gastric microbiome contributes to the development and progression of GC may improve GC’s prevention, diagnosis, and treatment. In this review, we discuss the current knowledge about changes in gastric microbial composition in GC patients, their role in carcinogenesis, the possible therapeutic role of the gastric microbiome, and its implications for current GC therapy.

## 1. Introduction

Gastric cancer (GC) is the fifth most common cancer worldwide, with a higher incidence in Asia, Eastern Europe, Central, and South America, and lower in North America and Africa [1,2]. Every part of the organ may be affected (from gastroesophageal junction to pylorus), with the *H. pylori* has been identified as the infectious agent most related to cancer development, especially of non-cardia GC (89%) [3,4]. In recent years, with the improvement of the investigations regarding the gastric microbiome, the contribution of other bacteria to gastric carcinogenesis has been identified [5,6,7]. However, detecting infective agents with the carcinogenic potential of *H. pylori* is challenging. A small number of studied cases, methodological variability, and population diversity represent some of the reasons limiting this issue’s knowledge; nevertheless, whether the microbial changes observed in GC are a promoter or a consequence of the histologic progression through the precancerous cascade [8].

On the other hand, there are diversities in the various populations worldwide regarding chronic *H. pylori* infection and the topography of gastric cancer [9]. Moreover, chronic inflammation may progress from atrophic gastritis to intestinal metaplasia, dysplasia, and gastric adenocarcinoma. [10]. Studies of the relationship between the microbiome and gastric cancer focused on hypochlorhydria, which is almost associated with atrophic gastritis, even due to the difficulties in culturing commensal microorganisms residing in the stomach. In this context, the number of microbes capable of surviving in the stomach was limited [11,12,13]. Recent advances in PCR techniques and metagenomics made possible the accurate study of the microbiome. As a result, an increased investigation into the relationship between the gastric microbiome and gastric cancer is considered [14].

Since conventional treatments, which include surgery, chemotherapy, and radiological therapy, do not offer a considerable improvement in the survival of these patients, new treatment strategies are mandatory. Bacteria have long been considered promoters of carcinogenesis, but their antineoplastic properties have only recently been discovered. As recent studies have reported, some bacteria may enhance the effect of some traditional antineoplastic drugs and immunotherapies.

This review will discuss the role of the microbiome in carcinogenesis and gastric cancer treatment, focusing on possible clinical implications.

## 2. Bacteria and Carcinogenesis

It has already been reported that some cancers are associated with infection by specific bacteria [15]. This is true, for example, of Helicobacter pylori, which has been found to contribute to the development of GC and MALT lymphoma through chronic inflammation [16,17]. Other examples encompass Fusobacterium nucleatum associated with colorectal cancer, Salmonella typhimurium related to cancer of the gallbladder, *Mycoplasma* spp. correlated with lung cancer, *Clostridia* spp. and Ruminococcaceae linked with breast cancer, and Chlamydia trachomatis and Mycoplasma genitalium associated with ovarian cancer (Figure 1) [18,19,20,21].

The infection by *H. pylori* can induce DNA mutation by disrupting double-strand DNA breaks, minimizing mismatch repair genes and aberrant DNA methylation, and maximizing activation-induced cytidine deaminase expression [22,23].

Three molecules are critical for the potential oncogenic burden of *H. pylori*: the cytotoxin-associated gene A (CagA) protein, the vacuolating cytotoxin A (VacA) protein, and the outer inflammatory protein A (OipA) [24,25,26]. The CagA penetrates the gastric epithelial cells only when T4SS is bound with α5β1 integrin. Afterward, the translocated CagA stimulates the production of IL-8 and provokes the dysregulation of the cell signaling via two pathways, a phosphorylation-dependent and phosphorylation-independent [27,28]. In the phosphorylated pathway, the phosphorylated CagA protein with src homology phosphatase 2 (SHP2)) along with the growth factor receptor-bound protein 2 (Grb2)48 activate oncoproteins such as extracellular signal-regulated kinase (ERK), mitogen-activated protein kinase (MAPK), and CT10 (chicken tumor virus No. 10) regulator of the kinase (Crk) Crk/Crk-linker (Crk-L) signaling pathways, that lead to irregulated cell proliferation and the development of GC [27,29,30].

In the unphosphorylated pathway, the CagA activates NF-κB, rat sarcoma (RAS), wingless integration-1 (Wnt)/β-catenin64, and phosphatidylinositol 3-kinase/Akt pathways by interacting with Met, E-cadherin, Grb2, and the polarity-regulating kinase partitioning-defective 1b (Par1b) signaling proteins. These activations cause morphological cell changes, irregular proliferation, and GC (Figure 2) [27,31,32].

## 3. Gastric Microbiome

It has been demonstrated that the community of the gastric microbiome comprises a population of its own. For this purpose, the diversity and composition of the microbial community are determined by analyzing its16S rRNA gene sequence.

First, Schulz et al. [33] reported that *H. pylori*-infected and non-infected individuals exhibited significant differences between the gastric luminal and gastric mucosal bacterial communities and no significant differences in the structure or composition of the oral microbial communities. Recent studies have provided that the gastric microbiome is selected by the gastric environment itself [34,35,36]. In patients without *H. pylori* infection or in treatment with proton pump inhibitors (PPI), luminal gastric microbiota consists mainly of Firmicutes, Bacteroidetes, Actinobacteria, and Proteobacteria. In contrast to the lower intestinal tract (and feces), where Firmicutes are dominant, followed by Bacteroidetes [37]. Similarly, regarding mucosal microbiota in patients non-infected by *H. pylori*, acid-resistant species, such as Veillonella, Lactobacillus, and Clostridium are the predominant species, while in *H. pylori*-infected patients, Streptococcus, Neisseria, Staphylococcus, and Roche were also identified [38].

A study that analyzed the mucosal microbiome along the gastrointestinal (GI) tract showed significant diversities between the upper and the lower GI tract, with the upper GI microbiome less affluent than the lower GI microbiome. Vuik et al. [34] found that Firmicutes, Proteobacteria, and Bacteroidetes were the main phyla. However, the upper GI is populated mainly by Proteobacteria and Firmicutes, while in the lower GI, Firmicutes and Bacteroidetes are the dominating phyla. More specifically, researchers have found that Veillonellaceae, Pseudomonadaceae, Streptococcaceae, Prevotellaceae, and Helicobacteraceae were the most prevalent bacteria of the gastric antrum. Other studies reported similar results from analyses carried out using culturomics with MALDI-TOF combined with 16S rRNA gene sequencing [35].

However, studies concerning transcriptional microbiome identification from saliva, upper (body and antrum of the stomach and duodenum), lower GI tract (terminal ileum, ascending and descending colon), and feces reported high heterogeneity regarding the structure but not the sites between upper and lower GI tract. Other authors used different technical approaches for bacteria identification, but they agreed that Firmicutes, Bacteroidetes, and Proteobacteria dominated, whereas Streptococcus, Pseudomonas, Prevotella, and Helicobacter were the most abundant genera in the stomach [34,35].

## 4. Gastric Microbiota in Gastric Carcinogenesis

Recently, two large studies comparing the gastric microbiome in groups of patients with different histological stages of cascading gastric carcinogenesis have been carried out [5,39]. Ferreira et al. [5] from Portugal showed a significant decrease in microbial diversity from chronic gastritis (81 patients) to gastric cancer (54 patients). The most apparent differences in this group of patients are the decrease in the abundance of Helicobacter and Neisseria and the growth of Citrobacter Lactobacillus and Clostridium in the gastric cancer microbiome (Table 1). Two groups of patients were distinguished in receiver operating characteristic analysis (ROC) when several abundant taxes were combined in a microbial dysbiosis index.

The gastric cancer microbiome displayed an increase in the functional characteristics of nitrosing, an aspect compatible with the microbial community with increasing genotoxic potential. The results were confirmed by the quantitative reaction of the polymerase chain PCR and confirmed in groups of different geographical origins.

In a study from China involving 81 patients (21 superficial gastritis, 23 atrophic gastritis, 17 intestinal metaplasias, and 20 gastric cancers), Coker et al. [39] demonstrated that there was a significant decrease in gastric microbial abundance in patients with intestinal metaplasia and gastric cancer, compared to patients with superficial gastritis. They found that *Peptostreptococcus stomatis*, *Streptococcus anginosus*, *Parvimonas micra*, *Slackiaexigua,* and *Dialister pneumosintes* were enriched in the gastric microbiota cancer patients, central to the gastric cancer microbial interaction network, and able to distinguish gastric cancer from surface gastritis in the ROC analysis (Table 1). All these results were successfully demonstrated in the consistency of Inner Mongolian patients.

Liu et al. [42] have shown that the gastric microbiome of cancer patients differs between different gastric microhabitats. In a specific study of 276 Chinese gastric cancer patients, microbial richness decreased significantly from normal gastric tissue in the area surrounding the tumor to tumor tissue itself. While in tumor tissues, there was an enrichment of *Prevotella melaninogenica*, *Streptococcus anginosus,* and *Propionibacterium acnes*, and a decrease in the abundance of *H. pylori*, in regular and peritumoral areas, *H. pylori* affected the overall structure of the microbiome (Table 1). The microbiome of three gastric microhabitats also had different bacterial correlation networks and functions. Specifically, the microbiome at the tumor site showed a less complex network of interactions than that of the normal peritumoral and microhabitats and was significantly enriched in the predicted functional genes involved in nucleotic transport and metabolism, as well as in amino acid transport and metabolism.

It remains to be elucidated whether changes in the gastric microbiome in different stomach microhabitats have a role in gastric carcinogenesis or are a consequence of tumor evolution.

Hu et al. [40] conducted a whole metagenome sequencing (WMS) investigation in the microbiome of gastric lavage samples from six gastric cancer patients and five patients with superficial gastritis and identified compositional and functional differences between clinical diagnoses. Microbial enrichment decreased significantly in the gastric cancer microbiome, characterized by the enrichment of commensal microorganisms or opportunistic pathogens of the oral cavity and the metabolic pathways of lipopolysaccharide and amino acid biosynthesis (Table 1). So far, no report has certainly directed WMS to gastric mucosal samples. In fact, in this type of sample, the amount of host DNA (>96%) significantly reduces the sensitivity of WMS to the profile of the microbiome, especially to detect fragile and scarce types of bacteria [41].

Animal models were important in showing the importance of the microbiome in gastric carcinogenesis. The transgenic insulin-gastrin mouse model (INS-GAS) develops gastric intraepithelial neoplasia (GIN) seven months after *H. pylori* infection. Lofgren et al. [43] demonstrated that compared to *H. pylori*-infected INS-GAS mice that harbor a complex gastric microbiome, the INS-GAS microbiome without germs infected with *H. pylori* had less heavy gastric lesions and delayed the onset of GIN (Table 1). The importance of the microbiome in promoting gastric neoplasia was also demonstrated in the K19-Wnt1/C2mE (Gan) mouse model of gastric carcinogenesis, which has simultaneous action of the Wnt and PGE22 pathways. Oshima et al. [44] showed that unlike Gan mice raised in conditions without specific pathogens, which develop large gastric tumors at 55 weeks of age, gastric tumorigenesis is significantly suppressed in Gan mice grown in germ-free conditions. The authors also indicated that re-hosting commensal bacteria in mice free of Gan germs or infection with Helicobacter felis led to the development of gastric tumors (Table 1). It was suggested that colonizing the stomach with commensal bacteria from other parts of the GI region could promote gastric carcinogenesis associated with *H. pylori*. Using the INS-GAS mouse model, Lertpiriyapong et al. [45] described the colonization of *H. pylori*-infected mice with a limited gut microbiome (*Clostridium* spp., *Lactobacillus murinus,* and *Bacteroides* spp.) summarized the histopathological results and the incidence of GIN observed in *H. pylori*-infected INS-GAS mice that harbor a complex microbiome. Although animal models are fundamental models of in vivo disease observation, differences in microbiome composition can be an essential source of experimental variability.

Ge et al. [46] indicated that C57BL/6 (B6) mice from various providers show significant differences in the structure of microbial communities along the gastrointestinal tract, which probably explains the different responses to *H. pylori* infection. In H-pylori infection, there were several efficiencies of gastric colonization and distinct influence on the structures of microbial stomach communities, colons, and feces of B6 mice from different providers. The pathological reactions and immunology status that the animals of the two sellers developed in response to *H. pylori* infection were also different. These results are in line with those of Velázquez et al. [47], who recently reported that differences in microbiome composition in genetically similar mice from various commercial suppliers were responsible for divergent phenotypes of susceptibility to salmonella infection. These results emphasize the importance of the microbiome in the reproducibility of animal experiments.

## 5. Gut Microbiota in Gastric Cancer

The gut microbiome affects many types of cancer and can also affect GC carcinogenesis and the response and prognosis of gastric cancer treatment. The importance of the gut microbiome for the interactions between cancer and immunity is gaining appeal, along with a further introduction of immunotherapies into clinical practice.

Neoplasms of the gastrointestinal (GI) tract have differences in their microbiome. Feces of patients with GC contain a high level of Enterobacteriaceae, whereas feces of patients with colon and rectal cancer possess lower levels of Lactobacillaceae and Bifidobacteriaceae, respectively [48].

It has been demonstrated that surgery for GC influenced the postoperative composition of the gut microbiome. In particular, patients who underwent distal gastrectomy had Escherichia/Shigella, Veillonella, and Clostridium XVIII in great abundance and Bacteroides in lower quantity [49]. Studies reported that fecal microbiota alterations, especially in the dominant finales of Bacteroides, Firmicutes, and Proteobacteria, may be implicated in the progression of gastric lesions associated with *H. pylori* infection [50]. Furthermore, distal gastrectomy may alter the microbiome of the oral cavity, with an increase of Escherichia-Shigella, Enterococcus, Streptococcus, and other typical bacteria (Veillonella, Oribacterium, and Migibacterium) [51]. Finally, changes in gut microbiome composition after bariatric surgery have been linked with long-term metabolic consequences [52,53].

New therapies either for oesophagic cancer or GC have been developed recently. The FDA has approved several targeted therapeutic agents. Pembrolizumab and Nivolumab are based on testing for MSI by PCR or NGS/MMR by IHC, PD-L1 immunohistochemical expression, or high tumor mutational burden (TMB) by NGS. It seems that gut microbiome target programmed cell death 1 (PD-1) may interfere with primary resistance to immune control inhibitors (ICI). Animal studies have shown that germ-free or antibiotic-treated mice had better antitumor effects of PD-1 blockade after microbiota transplantation from feces by cancer patients who responded to ICI.

In addition, metagenomic studies of samples from the stool of patients with GC showed a positive correlation between clinical ICI responses and the relative abundance of *Akermansia muciniphila* (*A. muciniphila*). Oral supplementation with *A. muciniphila* after non-responders FMT continues to restore the effectiveness of PD-1 blockade [54].

## 6. Gut Microbiota in Gastric Carcinogenesis

It is reliable that GC is associated not only with gastric dysbiosis but also with intestinal dysbiosis. The gut microbiota could be applied in gastric carcinogenesis through immunomodulation, but it may also affect the effectiveness of chemotherapy in patients with GC.

Qi et al. [55] showed intestinal dysbiosis in patients with GC and correlated them with peripheral cellular immunity. Venn diagram analyses showed 35 unique operational taxonomic units (OTUs) in healthy subjects (*n* = 88) and 240 unique OTUs in GC patients (*n* = 116). Two enterotypes were identified in this population; Enterotype 1 was dominated by Bacteroides, and enterotype 2 by Prevotella 9. However, enterotype distribution displayed no significant difference between GC patients and controls. Increased richness decreased butyrate-producing bacteria, and enrichment of 12 genera (such as Lactobacillus, Escherichia, and Klebsiella) was reserved in GC patients compared to subjects. Random forest analysis showed that the combination of Lactobacillus, Tyzzerella 3, Veillonella, Streptococcus, and Lachnospira was sufficient to distinguish GC patients from healthy subjects (Table 2). In addition, CD3+ T cell counts were linked to the relative abundance of Lactobacillus and Streptococcus, while CD3+ T cells, CD4+ T cells, and NK cells were linked to Lachnospiraceae.

Other authors evaluated the impact of gastrointestinal hormones on inflammation and gut microbiota in Chinese patients with GC. Serum levels of gastrin-17, pepsinogen II, IL-6, and IL-17 increased in patients with GC and are related to disease severity. After chemotherapy with FOLFOX4 (i.e., oxaliplatin/5-fluorouracil/leucovorin), gastrin-17 and pepsinogen II remained constant, while IL-6 levels decreased and pepsinogen I increased in these patients. The gut microbiota was then studied using fecal samples and standard culture-based identification methods. Bifidobacterium, Lactobacillus, and Bacilli were less abundant in gastric cancer patients’ intestines than their controls, and levels were refilled after chemotherapy with FOLFOX4. *Escherichia coli*, Staphylococcus, Enterococcus, and Peptostreptococcus were more productive in the gut of patients with GC than controls, and their levels decreased after chemotherapy with FOLFOX 4. Interestingly, gastrin-17 was negatively associated with the relative abundance of Bifidobacterium and Lactobacillus in the intestines of patients with gastric cancer (Table 2) [56].

Liang et al. [49] analyzed fecal samples from GC patients before and after radical distal gastrectomy (*n* = 6) to explore changes in the gut microbiota of GC patients in the perioperative period. Surgery had a limited effect on diversity; However, the gut microbiota composition was significantly influenced, indicating a higher relative abundance of Akkermansia, Escherichia/Shigella, Lactobacillus, and Dialister. In addition, when the gut microbiota of patients with GC (*n* = 20) was compared with those of healthy controls (*n* = 22), a higher relative abundance of Escherichia/Shigella, Veillonella, and Clostridium XVIII and a lower abundance of Bacteroides were identified (Table 2). Interestingly, the genus Helicobacter showed low plenty (<1%) in fecal GC samples and patients they control.

Yu et al. [57] provided mechanistic support for the role of the gut microbiome in gastric carcinogenesis using a chemically induced GC mouse model. Gastric carcinogenesis was induced in male mice without Wistar pathogens (SPF) using a combination of N-met-yl-N’-nitro-N-nitrosoguanidine (MNNG), sodium salicylate, irregular fasting, and ranitidine, which led to an increase in richness (ACE and Chao indices) but decreased the diversity (Shannon index) of the gut microbiota. The gut microbiota composition also varied between normal and pathological states (non-atrophic gastritis, chronic atrophic gastritis, precancerous lesions, and GC). Mice with precancerous lesions had the highest ratio of Firmicutes/Bacteroidetes. In contrast, mice with GC had a high abundance of Lactobacillus, Bifidobacterium, and Escherichia-Shigella and a low abundance of Lachnospiraceae and Ruminococcaceae in their intestines, partly supporting previous results in human studies (Table 2) [49,55,56].

## 7. Surgery for Gastric Cancer and Microbiome

Gastrectomy represents the cornerstone of the treatment for GC. It has been found that surgery improves survival rates somehow, but nowadays, effective treatment for GC is still lacking. The cornerstone of curative therapy for GC is surgical resection with lymphadenectomy. Nearly 50% of GC patients may undergo resection with curative intent. After surgery, the five-year survival rate is around 45%, with perioperative chemotherapy improving that rate by about 10% [58].

The potential postsurgical benefits of probiotics (4 strains: *Lactobacillus plantarum* MH-301 (CGMCC NO. 18618), *L. rhamnosus* LGG-18 (CGMCC NR. 14007), *L. acidophilus* and *Bifidobacterium animalis* subsp. lactis LPL-RH (CGMCC NO. 4599) has been evaluated in a clinical study. The authors concluded that probiotics could restore the homeostasis in gut microbiota, reduce inflammation, maintain the intestinal barrier and immunity, and benefit the recovery and prognosis [59].

Although gastrectomy aims to achieve an R0 resection which means radical resection of the primary tumor and the lymph nodes, it has been indicated that surgery for GC reported changes in the microbiome of patients with gastric cancer [60]. Tseng et al. [61] found modifications in the gastric microbiome’s diversity and community composition in patients who underwent surgery for GC.

Lin et al. [62] analyze the fecal microbiome in patients after gastrectomy. They indicated that patients with total gastrectomy, especially those who underwent a Roux en Y anastomosis, were implausible to have type II diabetes and metabolic syndrome than controls. Nonetheless, the post-surgery consequences that microbiome changes bring about on patients’ condition, particularly after total gastrectomy, remain unclear.

There is increasing evidence of a possible link between the gut microbiome and postoperative outcomes. For example, patients with GC who underwent gastrectomy might be at increased risk of developing metachronous cancer, including colorectal cancer (CRC) [63,64]. Previous studies have reported the incidence of metachronous cancers in such patients. Eom et al. found that the three most common sites of metachronous cancer were colorectum (20.8%), lung (11.9%), and liver (11.3%). Ikeda et al. demonstrated that colorectal cancer was the most familiar metachronous cancer (32.6%), followed by lung cancer (28.4%) and liver cancer (8.4%). Lundegardh et al. indicated that the most typical type of metachronous cancer was colorectal cancer (19.9%), followed by lung cancer (6.1%) and kidney cancer (5.3%). Kim et al. displayed that the colorectum was the most frequent location of metachronous cancer (26.3%), followed by the lung (23.7%) and liver (18.4%) [65]. The mechanism for metachronous CRC after gastrectomy is not like sporadic cancer. Patients who underwent gastrectomy exhibited higher CRC-enriched microbes (*F. nucleatum* and *Atopobium parvulum*) than those who did not. *F. nucleatum* has been suggested to mediate the early steps of carcinogenesis through FadA adhesion to the epithelium, activation of β-catenin signaling, and infiltration of myeloid cells into the tumor microenvironment. The enrichment of *F. nucleatum* and *A. parvulum* in these patients has been correlated with multiple polypoid adenomas and intramucosal carcinoma. Enrichment of these species suggests a form of dysbiosis, leading to CRC development after gastrectomy [66].

Gastric resection and reconstruction lead to changes in oxygen availability, intestinal pH, food transit time, intestinal motility, and hormonal activity, affecting the microbiome and fecal metabolism. Patients who underwent gastrectomy display different microbiome composition and metabolite profiles compared to the control group. The gastrectomy group exhibited a lower dissimilarity index than the control counterparts, indicating more remarkable species similarity between post-gastrectomy individuals [67,68].

As reported previously, there is more incredible richness and variety of species in post-gastrectomy patients [62,69,70,71]. Significant changes in the intestinal environment occur after gastrectomy and reconstruction, leading to the growth of certain species. The concentration of oxygen in the gut after gastrectomy is higher, resulting in aerobic and anaerobic microbial growth [69,70,71,72]. Various aerobics (Streptococcus and Enterococcus) and anaerobes (Escherichia, Enterobacter, and Streptococcus) were higher in patients who underwent gastrectomy than in controls. Besides, oral microbes might migrate to the gut [73].

## 8. Microbiome and Gastric Cancer Treatment

Conventional treatments for GC include surgery, chemotherapy, and radiotherapy. However, the benefit in terms of long-term survival is poor. In such a context, new treatment strategies are needed [74]. Recent studies attribute anti-neoplastic characteristics to the bacteria. The impact of the gastric microbiome on the treatment of GC has not yet been explored. Moreover, studies have highlighted microbes’ importance and potential implications for disease recovery [75].

It has been shown that the microbiome can strengthen the immune system against cancer. For example, cyclophosphamide (CTK) affects gut microbes. These microbes promote the creation of immune cells, which in turn improve the efficacy of CTK [76].

Microbes have been found to promote carcinogenesis by promoting inflammation. This stimulating response can also improve cancer treatments. Oxaliplatin, cisplatin, and CpG-oligonucleotide immunotherapy depend on inflammation. Antibiotic-treated mice (which killed the gut microbiome) did not respond as well to platinum chemotherapy or CpG-oligonucleotide immunotherapy as mice with intact gut microbes [77]. In antibiotics-treated or germ-free mice, tumor-infiltrating myeloid-derived cells are poorly cured by therapy, leading to decreased cytokine production and tumor necrosis after CpG-oligonucleotide treatment and lacking production of reactive oxygen species and cytotoxicity after chemotherapy [78]. Candida species may induce myeloid cells to infiltrate tumor sites and increase the production of pro-inflammatory cytokines. This could result from the intestinal dysbiosis caused by antibiotics, leading to an increase in the plasma concentration of prostaglandin E2 and the polarization of M2 macrophages in the lungs, leading to a rise in allergic respiratory irritation [79]. These conditions may cause rapid hemorrhagic necrosis in treated tumors [80]. Recent studies have demonstrated that responses to anti-PD1 immunotherapies in melanoma patients are linked with a high diversity and abundance of Ruminococcaceae/Fecalibacterium [81]. Bacteroides thetaiotaomicron and Bacteroides fragilis have been confirmed to improve cancer immunity from lymphocyte-4-associated antigen antibodies (CTLA-4) by modulating dendritic cells. The resultant mucosal lesions and Bacteroides translocation are necessary to activate the immune system and create a more appropriate antitumor environment [82].

Although microbiota shows beneficial effects, for example, when wide-range antibiotics are administered by promoting dysbiosis, sometimes the microbiota can also influence side effects. For example, irinotecan (CTP-11) brings about severe diarrhea created by its complex activation and subsequent metabolism. SN38 produced by carboxylesterases is glucuronidated in the liver by diphosphate uridine (UDP)-glucuronosyltransferase enzymes to form inactive SN-38G, which is excreted through the biliary ducts into the gastrointestinal tract. When it reaches the intestine, it is activated again by some diarrhea-promoting bacteria that limit the dose through damage to the epithelial barrier [83].

Therefore, understanding how bacteria change and their relationship to the immune system are the keys to making the gut microbiome a holistic focal point for cancer development while improving clinical and therapeutic protocols.

## 9. Chemotherapy

Commonly used chemotherapy drugs for GC encompass fluorouracil, gemcitabine, capecitabine, oxaliplatin, and irinotecan. The anticancer activity of fluorouracil (as well as its proruminal capecitabine) and gemcitabine (pyrimidine antimetolite) causes disruption of DNA synthesis, DNA damage, and ultimately induction of apoptosis in cancer cells. Fusobacterium nucleatum activates cancer cells in autophagy, preventing tumor cell apoptosis, and increases the amount of BIRC3, a protein that directly hinders apoptosis by connecting caspases [64,81]. In colorectal cancer, Fusobacterium nucleatum interferes with molecules such as Toll-like receptors and microRNA in order to control resistance to oxaliplatin [84]. Resistance to gemcitabine is brought about by its break down into an inactive metabolite (2’,2’-difluorodeoxyuridine) due to cytidine deaminase. The long isoform of cytidine deaminase is expressed by several gut bacteria, mainly Gammaproteobacteria [85].

Another point of current research is focused on using gut microbiota composition as a predictor of response to treatment, as it is with capecitabine in CRC [77]. The antineoplastic effect of oxaliplatin is expressed by preventing DNA replication due to intra- and inter-chain cross-links and promoting ROS formation that damages DNA and causes apoptosis of tumor cells [86]. Similarly, Fusobacterium nucleatum promotes chemoresistance of oxaliplatin by activating autophagy in tumor cells [86]. Studies in animals have shown that antibiotics limit the antineoplastic effect of oxaliplatin by preventing tumor infiltration of ROS-producing leukocytes [78].

Specific types of gut bacteria have been linked with resistance to cancer treatments. In patients with colorectal cancer who showed drug resistance, an increase in the nucleus of Fusobacterium nucleatum in the gut has been found. This bacterium seems to block apoptosis in neoplastic cells and cause autophagy [87].

## 10. Microbiome and Cancer Immunotherapy

It has been reported that advanced melanoma can be successfully treated with immunotherapy. In the face of this success, its application has been suggested for other types of cancer, such as metastatic melanoma. Up to 60% of patients with this disease can now achieve stable trade-offs thanks to immunotherapy [88]. Three recent studies suggest an association between the gut microbiome and the patient’s response [54,81,89]. These studies revealed differences in gut microbiome composition of responders and non-responders. However, all three studies disagreed on specific microbes that were important for the success of immunotherapy [90]. Moreover, since there was no agreement regarding which commensal microbes matter most, the gut microbiome mechanisms that affect immunotherapies remain unclear [91].

Microbiome, in response to inflammatory signals, expresses its antineoplastic properties by promoting the production of interferon (IFN)-γ and the large B enzyme of CD4 and CD8 T cells and the recruitment of antitumor macrophages. Studies in melanoma models have found that It has been found that checkpoint inhibitors are less effective in microbe-free and antibiotic-treated mice than in mice with intact gut microbiome [92]. Furthermore, it has been suggested that specific bacteria, such as Bacteroidetes thetaiotaomicron and *Faecalibacterium prausnitzii* (*F. prausnitzii*), can be used in probiotic treatments to increase the effectiveness of checkpoint inhibitors [93].

A recent study demonstrated that gut bacteria produce short-chain fatty acids (SCFAs), which may act as promoters for the memory potential of antigen-activated CD8+ T cells. *F. prausnitzii* has been identified as one of those (leading clostridial producers of SCFAs, including butyrate) in patients who responded favorably to control blockade. It seems that gut microbes such as *F. prausnitzii* were helpful in cancer immunotherapies by promoting different phenotypes based on various environmental and host-particular factors [94]. An explanation of how SCFA-producing clostridia impacts immune control blockade’s effectiveness lies in that some gut microbiome members may have a common functional outcome (SCFA, butyrate production), whereas metabolites they produce are functionally important. Moreover, since immunotherapy may affect the gut microbiome, its study would be imperative [95].

A study in patients with metastatic melanoma treated with ipilimumab showed gut bacteria in the microbiome of patients with pre-colitis who later developed colitis [94]. These patients had fewer Bacteroidetes Phylum bacteria and lower expression of genes involved in polyamine transport and vitamin B biosynthesis. These findings could help explain why some patients develop colitis, which could lead to miscarriage, and reduce the risk of inflammatory complications from cancer immunotherapies. Immunotherapies can cause intestinal inflammation that changes the microbiome’s composition, causing a feedback loop that can lead to more complications [96].

## 11. Relation of Gut Microbiome and Immune Response in Gastric Cancer

The intestinal microbiome is essential for developing GC and might affect the response to immunotherapies. Intestinal bacteria induce an antineoplastic immune response through several mechanisms. They promote the T-cell response to bacterial antigens that can cross-react with tumor antigens. Moreover, they recognize tumor-specific antigens by recognizing immune or anti-inflammatory metabolites mediating systemic effects in the host [97].

Specific T-cell receptors are activated by the peptide or lipid composition of bacteria. Das et al. [98] found that Helicobacter pylori could increase the expression of gastric epithelial factor PD-L1. The expression of PD-L1 caused a deceleration in the proliferation of isolated CD4 + T cells in the blood. On the other hand, anti-PD-L1 antibodies ruled out the inhibitory effect of PD-L1 [99]. Wu et al. [100] demonstrated that PD-L1 expression in primary human gastric epithelial cells was strongly enhanced by *H. pylori* infection and activated T cells. Moreover, PD-L1 expression in gastric epithelial cells significantly induced apoptosis of T cells. Thus, *H. pylori* infection may inhibit circulating T-cells, including tumor-specific T-cells.

Liu et al. [101] displayed that PD-L1 was expressed in 59.3% of GC patients and was correlated with positive *H. pylori* status, high microsatellite instability, and Epstein-Barr virus positivity. These results indicate that some patients with GC and *H. pylori* infection might benefit from anti-PD-1/PD-L1 therapy.

Vétizou et al. [82] exhibited that different Bacteroids species are responsible for the antitumor effects of CTLA-4 blockade. Studies conducted in animals and patients showed that the efficacy of CTLA-4 blockade depends on T cell responses specific for *B. thetaiotaomicron* or *B. fragilis*. The CTLA-4 blockade displayed no impact on tumors in antibiotic-treated or germ-free mice. Gavage with *B. fragilis*, immunization with *B. fragilis* polysaccharides, or adoptive transfer of specific *B. fragilis* T cells may defeat this imperfection [102]. Bifidobacteria might provoke up-regulation of interferon type I related immune genes in antigen-presenting cells of secondary lymphoid organs [93]. In patients with advanced cancer, antibiotics inhibited the clinical benefit of immune blocking inhibitors. Furthermore, fecal microbiome transplantation (FMT) from responders to immune control inhibitors in sterile mice increased the antitumor effects of PD-1 blockade, while FMT from non-responders did not [102].

The gut microbiome affects host metabolism by generating small peptides that influence host immune metabolism. Spermidine and vitamin B6 produced in the gut can stimulate autophagy in remote areas of the body, triggering anti-cancer immune responses in the context of chemotherapy. SCFA produced by gut bacteria is detected by various cell types, including regulatory T cells expressing GPR41 or GPR43 receptors paired with the G protein. Bacteria-derived aldehyde dipeptides can mediate the inhibition of cathepsin L, thus interfering with the presentation of epithelial antigen or immune cells [103].

Since 1867, when Streptococcus pyogenes infection was reported to cause cancer readmission, it was proposed that the microbiome may have a role in cancer treatment [104]. Multiple mechanisms have been proposed regarding antineoplastic effects by which the bacteria could exert anticancer effects. Some examples are colonizing tumors, releasing substances, suppressing nutrients necessary for tumor metabolism, and spread, and increasing host immunity [105,106]. The ribosomal protein helicobacter pylori (HPRP)-A1 HPRP-A1 and its enantiomer HPRP-A2 show strong antimicrobial and anticancer activities. HPRP-A1 can break down the tumor cell membrane by KLA-mediated action and is therefore often used to facilitate the administration of other drugs to cancer cells, while HPRP-A2 induces cellular apoptosis in GC cells by increasing ROS production, activation of caspases-3, 8 and 9, reduce the potential of the mitochondrial membrane, and cell cycle arrest in phase G1 [107,108,109,110,111]. In addition, b-ieodoglucomides, a glycopeptide isolated from the Bacillus licheniformis, have been shown to have cytotoxic activity against GC cells [112]. Also, FW523-3, a lipopeptide isolated from Micromonosporachalcea, has been shown to prevent the spread of GC cells [113].

In a recent study (DELIVER trial: UMIN000030850), 501 patients with advanced GC were treated with nivolumab. The primary endpoint was the study of a genomic pathway of the gut microbiome as a predictive marker of nivolumab efficacy. Secondary endpoints included relationships between markers of microbiome and clinical outcomes. The analysis of the bacteria genome showed that Odoribacter and Veillonella were associated with tumor response to nivolumab, which means that bacterial invasion of epithelial cells pathway in the gut microbiome may become a novel biomarker for the treatment of advanced GC with nivolumab. Therefore, predictive biomarkers are needed for nivolumab treatment in gastric cancer. We sought to investigate whether genomic information in the gut microbiome will serve as predictors for nivolumab in advanced gastric cancer. Moreover, the gastric cancer-specific gut microbiome may predict response to immune checkpoint inhibitors [114].

In conclusion, the next steps in the research of the role of the gastric and gut microbiome on the therapies for the GC are expected to include extensive multi-central prospective studies conducted in humans for the identification of specific bacterial species and pathways, as well as changes in the microbiome-related with the progression of GC. Thus, it would be feasible that changes in the microbiome could be used to control disease progression. In addition, manipulation of the gastric microbiome, other than the eradication of *H. pylori*, would have the potential to represent a disease-modifying treatment.

## Figures and Tables

**Figure 1 cancers-14-02039-f001:**
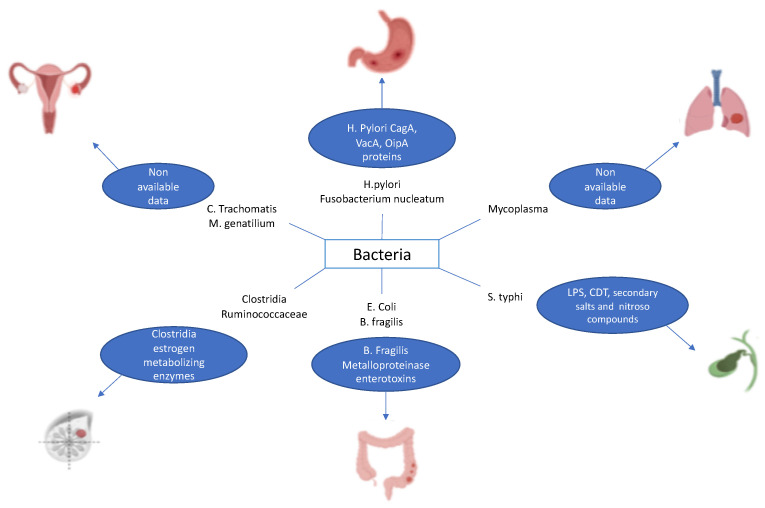
Examples of bacteria causing cancer.

**Figure 2 cancers-14-02039-f002:**
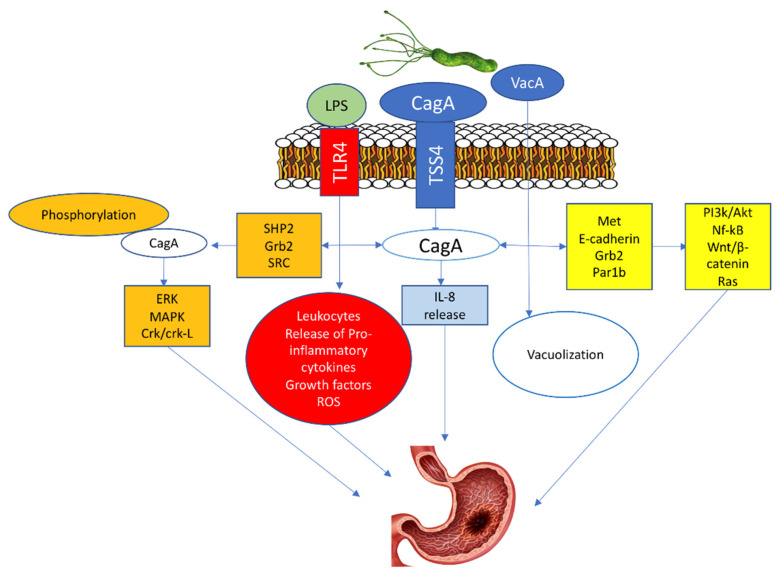
*H. pylori*-induced Gastric cancer.

**Table 1 cancers-14-02039-t001:** Gastric microbiota in gastric carcinogenesis.

S. No.	Bacteria Situation	References
1	Increased: Citrobacter; Lactobacillus; Clostridium Decreased: Helicobacter; Neisseria	[5]
2	Increased: *Peptostreptococcus stomatis*; *Streptococcus anginosus*; *Parvominas micra*; Slakiaexigua; *Dialister pneumosintes*	[37,38]
3	Increased: *Prevotella melanogenica*; *Streptococcus anginosus*; *Propionibacterium acnes* Decreased: *H. pylori*	[37,38]
4	Decreased microbiome	[39,40,41]

**Table 2 cancers-14-02039-t002:** Gut microbiome in gastric carcinogenesis.

S. No.	Bacteria Situation	References
1	Increased: Lactobacillus; *Escherichia coli*; Klebsiella; Tyzzerella_3; Veillonella; Streptococcus; Lachnospira	[55,56]
2	Increased: *Escherichia coli*; Staphylococcus; Enterococcus; Peptostreptococcus Decreased: Bifidobacterium; Lactobacillus; Bacilli	[56,57]
3	Increased: Akkermansia; Escherichia; Shigella; Lactobacillus; Dialister Decreased: Bacteroides; Helicobacter	[49,56,57]
4	Increased: Bifidobacterium; Lactobacillus; Escherichia; Shigella Decreased: Lachnospira, Ruminococcus	[49,56,57,58]

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
