# Peer review of "The Implication of Gastric Microbiome in the Treatment of Gastric Cancer"

_cancers, 2022, doi:10.3390/cancers14082039_

Round 1
Reviewer 1 Report
The manuscript has improved compared to the previous submission.
Author Response
On behalf of all authors, I would like to thank the Reviewer for his valuable comments.
Following his instructions, the manuscript has been improved adequately.
Reviewer 2 Report
The authors have revised their manuscript according to the comments and suggestions are given to them in the earlier review process.
However, the English language still needs to be checked by a native speaker.
Author Response
On behalf of all authors, I would like to thank the Reviewer for his valuable comments.
Following his instructions, the manuscript has been checked adequately regarding the English language.
This manuscript is a resubmission of an earlier submission. The following is a list of the peer review reports and author responses from that submission.
Round 1
Reviewer 1 Report
The authors review the role of gastric and gut microbiome in the development of gastric cancer and in its treatment.
- a figure showing how bacteria can promote carcinogenesis (such as secretion of metabolites, induction of inflammatory response, induction of DNA damage, and promotion of EMT etc) should be added to the text together with a detailed paragraph of the molecular mechanisms of action known so far (such as CagA for H. pylori, FadA for F.nucleatum, DnaK for Mycoplasma.......). A lot has been done both in vivo and in vitro and these information have important clinical implications;
- some sentences are truncated and do not make any sense. Please review the text;
- most of the references are a little bit outdated. Please add some updated references.
Author Response
Responses to the Reviewers Comments
First of all we would like to thank you for your valuable comments.
Following your instructions we aimed to accomplish your comments.
Comments and Suggestions for Authors
The authors review the role of gastric and gut microbiome in the development of gastric cancer and in its treatment.
• a figure showing how bacteria can promote carcinogenesis (such as secretion of metabolites, induction of inflammatory response, induction of DNA damage, and promotion of EMT etc) should be added to the text together with a detailed paragraph of the molecular mechanisms of action known so far (such as CagA for H. pylori, FadA for F.nucleatum, DnaK for Mycoplasma.......). A lot has been done both in vivo and in vitro and these information have important clinical implications;
Response
Since the topic of this article is focused on the implication of the gastric and gut microbiome on the treatment of gastric cancer, we believe that a such a figure would not add anything substantial to this topic.
However, if the Reviewer believes that such a figure would be necessary, then we will make it.
• some sentences are truncated and do not make any sense. Please review the text;
Response
The text has been reviewed according to your instructions.
• most of the references are a little bit outdated. Please add some updated references.
Response
References have been updated according to your instructions.

Reviewer 2 Report
In this review article, the authors have discussed the contribution of the gastric microbiome in the development of gastric cancer. they have mentioned the changes in gastric microbial composition in gastric cancer patients and their implication in gastric cancer therapy. Overall this article is poorly organized and lacks a lot of key information or a detailed explanation to clarify the role of the gastric microbiome in gastric cancer. I have several major concerns.
- recently two excellent and comprehensive reviews were published on the same topic [PMID: 35112169]. The current study seems to have no new information to add to the knowledge.
- Plagiarism detection showed 46% similarity from previously published studies of which the highest was from another review article [PMID: 32916853].
- The title of this study is a bit misleading, as very little information about the gut microbiota was given, and most of the information was about gastric microbiota.
- A lot of information seems unorganized and lacks clarity, I believe that the article was not properly proofread for grammar and organization and several authors worked separately on each section. Moreover, there are long complex sentences with no clear subject (lines206-209)
- Section 2, paragraph 2, lines 67- 72: details of bacterial communities found in luminal versus mucosa should be explained.
- Section 4, line 175: the section heading does not represent the information given below. the heading is very vague, the role of the gut microbiome in what situation? The information provided in this section is very broad and talks about probiotics, immune inhibitors, and non-small cell lung cancer.
- Line 195-203: this should be part of a separate section on the role of probiotics on gastric cancer.
- Line 211, positive or negative correlation?
- Lines 214-216: How is this information interesting? It is not clear how a study on lung cancer showing that patients responding to antibiotics and immunotherapy had worse prognoses have any link to the gut microbiome and gastric cancer. More details should have been given.
- Section5, para 1, lines 219-221: There is no link between the two sentences.
- Line 221: What is GS?
- Line 223: What is Qi and al...?
- Line 224: What is OTU?
- Line 275-276: Very strong claim without any supportive reference
- Lines 293-294: more details should be given, how much risk? what is the role of gastric microbiota in CRC?
- Line 313-314: Reference missing
- Line 478: what sort of association? more details should be given.
- Conclusions/future directions/clinical implications should be added at the end of the review.
Author Response
Responses to the Reviewers Comments
First of all we would like to thank you for your valuable comments.
Following your instructions we aimed to accomplish your comments.
- Comments and Suggestions for Authors
In this review article, the authors have discussed the contribution of the gastric microbiome in the development of gastric cancer. they have mentioned the changes in gastric microbial composition in gastric cancer patients and their implication in gastric cancer therapy. Overall this article is poorly organized and lacks a lot of key information or a detailed explanation to clarify the role of the gastric microbiome in gastric cancer. I have several major concerns.
- recently two excellent and comprehensive reviews were published on the same topic [PMID: 35112169]. The current study seems to have no new information to add to the knowledge.
Response
The aforementioned study [PMID: 35112169] treated the topic of microbiota AND gastric cancer.
Our study aimed to treat the implication of the microbiome in the TREATMENT of gastric cancer.
- Plagiarism detection showed 46% similarity from previously published studies of which the highest was from another review article [PMID: 32916853].
Response
We have tried to reduce the similarities.
- The title of this study is a bit misleading, as very little information about the gut microbiota was given, and most of the information was about gastric microbiota.
Response
The topic of this article was mainly focused on the implication of the gastric microbiome in the treatment of gastric cancer.
- A lot of information seems unorganized and lacks clarity, I believe that the article was not properly proofread for grammar and organization and several authors worked separately on each section. Moreover, there are long complex sentences with no clear subject (lines206-209)
Response
New therapies either for oesophagic cancer or GC has been developed recently. Several targeted therapeutic agentshave been approved by the FDA. Pembrolizumab/nivolumab are based on testing for MSI by PCR or NGS/MMR by IHC, PD-L1 immunohistochemical expression, or high tumor mutational burden (TMB) by NGS. It seems that gut microbiome target programmed cell death 1 (PD-1) may interfere with primary resistance to immune control inhibitors (ICI). Animal studies have shown that germ-free or antibiotic-treated mice had better antitumor effects of PD-1 blockade after microbiota transplantation from feces by cancer patients who responded to ICI.
- Section 2, paragraph 2, lines 67- 72: details of bacterial communities found in luminal versus mucosa should be explained.
Response
Ohno, H., Satoh-Takayama, N. Stomach microbiota, Helicobacter pylori, and group 2 innate lymphoid cells. Exp Mol Med 52, 1377–1382 (2020). doi.org/10.1038/s12276-020-00485-8
In patients without H.pylori infection or in treatment with proton pump inhibitors (PPI), it has been shown thatluminal gastric microbiota are mainly constituted by Firmicutes, Bacteroidetes, Actinobacteria, and Proteobacteria. In contrast to the lower intestinal tract (and feces), where Firmicutes are dominant, followed by Bacteroidetes.
Chen X, Xia C, Li Q, Jin L, Zheng L and Wu Z (2018) Comparisons Between Bacterial Communities in Mucosa in Patients With Gastric Antrum Ulcer and a Duodenal Ulcer. Front. Cell. Infect. Microbiol. 8:126. doi: 10.3389/fcimb.2018.00126
Similarly, regarding mucosal microbiota in patients non infected by H. pylori acid resistant species, such as Veillonella, Lactobacillus and Clostridium are the predominant species, while in H. pylori-infected patients, Streptococcus, Neisseria, Staphylococcus, and Roche were also identified.
- Section 4, line 175: the section heading does not represent the information given below. the heading is very vague, the role of the gut microbiome in what situation? The information provided in this section is very broad and talks about probiotics, immune inhibitors, and non-small cell lung cancer.
Response
The paragraph has been modificated appropriately according to your instructions.
- Line 195-203: this should be part of a separate section on the role of probiotics on gastric cancer.
Response
This section has been removed due to its minor contribution to the topic of this article.
- Line 211, positive or negative correlation?
Response
There was a positive correlation.
- Lines 214-216: How is this information interesting? It is not clear how a study on lung cancer showing that patients responding to antibiotics and immunotherapy had worse prognoses have any link to the gut microbiome and gastric cancer. More details should have been given.
Response
This sentence has been removed and replaced.
- Section5, para 1, lines 219-221: There is no link between the two sentences.
Response
Analogous corrections have been made to give the right sense.
- Line 221: What is GS?
Response
It should be GC (Gastric Cancer), instead of GS. Apposite corrections have been done.
- Line 223: What is Qi and al...?
Response
This is a study from Qi et al. The apposite correction has been done.
- Line 224: What is OTU?
Response
Operational taxonomic unit (OTU)
- Line 275-276: Very strong claim without any supportive reference
Response
We provide the apposite reference.
Khachfe HH, Salhab HA, Fares MY, Chahrour MA, and Faek Jamali FR. Ecancermedicalscience. 2021; 15: 1218. Published online 2021 Apr 8. doi: 10.3332/ecancer.2021.1218.
The cornerstone of curative treatment for GC is surgical resection with lymphadenectomy. Nearly 50% of GC patients may undergo resection with curative intent. Five-year survival rate after surgery is around 45%, with perioperative chemotherapy improving that rate by around 10%.
- Lines 293-294: more details should be given, how much risk? what is the role of gastric microbiota in CRC?
Response
Kim, C.; Chon, H.J.; Kang, B.; Kim, K.; Jeung, H.C.; Chung, H.C.; Noh, S.H.; Rha, S.Y.; et al. Prediction of metachronous multiple primary cancers following the curative resection of gastric cancer. BMC Cancer 2013 13, 394. doi:10.1186/1471-2407-13-394.
Previous studies have reported the incidence and clinical pattern of metachronous cancers in GC patients who underwent curative gastrectomy. Eom et al. reported colorectal (20.8%), lung (11.9%) and liver cancer (11.3%). In a similar study in Japan, Ikeda et al. reported colorectal cancer (32.6%), followed by lung (28.4%) and liver cancer (8.4%). Analogous results have been reported by Lundegardh et al. with colorectum (19.9%), lung (6.1%) and kidney (5.3%). In the study by Kim et al. the most common site of metachronous cancer was the colorectum (26.3%), followed by lung (23.7%) and liver (18.4%).
Erawijantari, P.P.; Mizutani, S.; Shiroma, H.; Shiba, S.; Nakajima, T.; Sakamoto, T.; Saito, Y.; Fukuda, S.; Yachida,S.; Yamada, T.; et al. Influence of gastrectomy for gastric cancer treatment on faecal microbiome and metabolome profiles. Gut 2020, 0, 1–12. doi:10.1136/gutjnl-2019-319188.
Although the mechanism for metachronous CRC after gastrectomy may not be similar to sporadic cancer, we observed higher abundances of several CRC-enriched microbes (eg, F. nucleatum and Atopobium parvulum) in the gastrectomy group. F. nucleatum has been suggested to mediate the early steps of carcinogenesis, through FadA adhesion to the epithelium, activation of β-catenin signalling and infiltration of myeloid cells into the tumour microenvironment. Thus, enrichment of F. nucleatum in the total gastrectomy group is worth noting.
In addition, A. parvulum has been associated with multiple polypoid adenomas and intramucosal carcinoma.Enrichment of these species suggests a form of dysbiosis, which can lead to CRC development after gastrectomy.
- Line 313-314: Reference missing.
Response
Joshi, S.S.; Badgwell, B.D. Current Treatment and Recent Progress in Gastric Cancer. CA Cancer J. Clin. 2021, 71, 264-279. doi: 10.3322/caac.21657.
- Line 478: what sort of association? more details should be given.
Response
Therefore, predictive biomarkers are needed for nivolumab treatment in gastric cancer. We sought to investigate whether genomic information in gut microbiome will serve as predictors for nivolumab in advanced gastric cancer.
- Conclusions/future directions/clinical implications should be added at the end of the review.
Response
The conclusions section has been added according to your instructions.

Round 2
Reviewer 2 Report
The authors only responded and modified the manuscript to minor and a few of the major comments.
Plagiarism detection still shows 45% similarity from previously published studies of which the highest was from another review article on a similar topic [PMID: 32916853]. Also the novel
The title was not changed even though the authors agreed that the study mainly focused on the implication of the gastric microbiome and not the Gut microbiome in the treatment of gastric cancer.